# Protein Maps for Durum Wheat Precision Harvest and Pasta Production

**DOI:** 10.3390/plants11223149

**Published:** 2022-11-17

**Authors:** Giovanna Visioli, Marta Lauro, Francesco Morari, Matteo Longo, Andrea Bresciani, Maria Ambrogina Pagani, Alessandra Marti, Gabriella Pasini

**Affiliations:** 1Department of Chemistry, Life Sciences and Environmental Sustainability, University of Parma, Parco Area delle Scienze 11/a, 43124 Parma, Italy; 2Department of Agronomy, Food, Natural Resources, Animals and the Environment, University of Padova, Viale dell’Università 16, 35020 Legnaro-Padua, Italy; 3Department of Food, Environmental, and Nutritional Sciences (DeFENS), Università degli Studi di Milano, Via G. Celoria 2, 20133 Milan, Italy

**Keywords:** precision farming, fertilization, gluten proteins, dough rheology, pasta quality

## Abstract

The quality traits of dough and dry pasta obtained from semolina (*Triticum durum* Desf. var. *Biensur*), harvested from a 13.6 ha field, subjected to variable-rate nitrogen (N) fertilization, were assessed to test site-specific pasta production for a short supply chain. Based on the grain quality spatial distribution, two distinct areas with protein content lower or higher than 13% were delineated and harvested selectively. The rheological properties of semolina samples obtained from those areas were evaluated. Furthermore, dry pasta was produced and characterized for its cooking behaviour and sensory characteristics. Semolina was demonstrated to have rheological characteristics (i.e., gluten aggregation time and energy, as evaluated by GlutoPeak test) positively related to the protein content as well as the related pasta, showing better cooking quality. These results are driven by the high amounts of gluten proteins, as well as by the glutenin/gliadin ratio, which are indicators of the technological quality of semolina. Overall, the results indicate that segregation of the grain with >13% of protein at harvest led to the production of semolina with a higher gluten protein content and a higher glutenin/gliadin ratio and, hence, to the production of pasta with better cooking quality. Therefore, site-specific pasta could be a potential asset for a short supply chain, aiming at improving traceability, as well as environmental and economic sustainability.

## 1. Introduction

Durum wheat (*Triticum turgidum* L. var. Durum) is an important crop used to produce pasta, couscous, bulgur and, in some areas of the world, various types of bread. Durum wheat occupies about 20–30 million hectares worldwide, distributed in many countries and representing 8% of world wheat production [1]. More than half of the total cultivation is found in the Mediterranean area, including Southern Europe, North Africa and Southwest Asia, where tetraploid wheats were domesticated between 10,000 and 15,000 BC [1,2]. Italy historically boasts a productive vocation for durum wheat, which plays a strategic role as a high-quality raw material for the pasta industry. In fact, Italy is the first producer of durum wheat in Europe and Italians have the highest per capita consumption rates of pasta in the world (23 kg/per capita/year) (http://www.internationalpasta.org, accessed on 5 May 2021). In this regard, the selection of genotypes of durum wheat with high production performance but that can commute the raw material in a high-quality product is fundamental for the Italian pasta market. Moreover, considering that pasta is more and more appreciated even outside Italy, this aspect is assuming a worldwide relevance.

The term “wheat quality” usually refers to the aptitude of the raw material to be transformed into a product that mainly depends on both quantity and characteristics of storage proteins that directly determine wheat’s market price and end-use value [3]. In this context, empiric rheology is considered a useful tool for assessing the technological quality of wheat. In the case of durum wheat, the Alveograph test is widely used to simulate the ability of semolina to withstand mechanical stress during pasta processing [4]. By measuring the pressure promoted by air insufflation that is necessary for the blowing—until breakage—of a dough disc, the test provides information on dough tenacity, extensibility and strength. About ten years ago, a new shear-based test was proposed for evaluating semolina quality based on its gluten aggregation kinetics [5]. Compared to conventional approaches, the GlutoPeak test presents several advantages: it requires less than 10 g of sample and less than 5 min for carrying out the analysis, it is easy to use and, above all, the operator influence is very low [4,5].

The rheological characteristics of wheat are determined by the amount and type of glutenins (GS) and gliadins (GLI). Glutenins, subdivided into high-molecular-weight glutenins (HMW-GS) and low-molecular-weight glutenins (LMW-GS), are organized in complex heteropolymers, which are stabilized by both intermolecular and intramolecular disulfide bonds. The combination of high-quality HMW-GS alleles and the high density of crosslinks between the shorter chains of LMW glutenins is known to be the main determinant of dough strength, which is a fundamental parameter for good-quality pasta [4,6]. Gliadins are mainly monomer proteins, divided in ω-, α/β- and γ-gliadins, and are responsible for dough extensibility. Thus, in addition to the total amount of gluten proteins, higher dough strength is related to the higher amounts of HMW-GS than LMW-GS polymers, although a high GS/GLI ratio is also important in determining the functional proprieties of gluten [7,8].

Although the total amount of gluten proteins is a genetically determined trait, evidence revealed that different managements and climatic conditions also influence it, as well as the relationship between the different gluten fractions [9,10,11]. In addition, the spatial variability in soil chemical and physical characteristics strongly affects the within-field grain quality, leading to a range of protein contents that justify the possibility of segregating grains during harvesting [12]. Morari et al. [13] demonstrated that grain segregation in durum wheat allows one to achieve the premium quality threshold (13.5%) recognized by the Italian cereal market, even in the critical cultivation zones of North-East Italy. In the experiment of Morari et al. [13], the average protein content of the field was roughly 12% in both the cropping seasons considered, but the sandy areas fertilized with high nitrogen input were eligible for the premium quality with protein content, reaching a maximum value of 15%. The same areas revealed high HMW/LMW-GS ratios, which allow for obtaining dough with good technological characteristics as well as pasta with good sensory properties [14].

Technology has been demonstrated to be almost mature to apply automatic grain segregation during harvesting [15] and produce “site-specific” pasta [14]. The bottleneck of a “site-specific” pasta supply chain stays in the definition of the minimum grain stock required in manufacturing pasta. Indeed, when zonal production is not sufficient, a multi-field or multi-farm approach should be pursued to guarantee an adequate tonnage of homogeneous grain stocks. New studies are, thus, necessary to evaluate how soil variability affects the grain quality and how this variability can lead to a site-specific short supply chain. In this context, the aims of this work were to: (i) map the quantity and quality of gluten proteins in the field according to the different soil-fertility zones; (ii) compare the protein maps obtained from the gluten protein contents measured from the laboratory tests with the two protein levels measured in the field; (iii) verify the technological quality of the semolina obtained from the segregating grains in the field and the cooking behavior and sensory characteristics of the related pasta.

## 2. Results and Discussion

### 2.1. Protein Maps

Figure 1 represents the map constructed from the data obtained from the total content of gluten proteins extracted from the kernels of 120 spikelets sampled in the different fertility zones in the field (Section 3, see Figure 6). The data on total gluten proteins extracted correlated well (r = 0.88) with the total N content measured on the same samples by NIR (Near-Infrared Spectrometer) (Appendix A). The highest amount of gluten proteins extracted was found in the medium-fertility zone (Figure 1), while lower amounts were found in the high-fertility zone (HFZ) and low-fertility zone (LMW).

Figure 2 reports the maps of the amount of the different gluten protein fractions (GLI, HMW-GS and LMW-GS) from the 120 spikelets sampled in the different fertility zones in the field (See Section 3 Figure 6). Different amounts of GLI, HMW-GS and LMW-GS were found according to the different fertility zones. Figure 2A showed that the highest amount of GLI was found in samples from the HFZ. The highest amount of glutenins was instead found in the medium-fertility zone (MFZ) (Figure 2B,C), in which the total GS fraction (HMW-GS+LMW-GS) and the LMW-GS fraction correlated well with total protein content by NIR (r = 0.59 and r = 0.57, respectively) (Appendix A), while lower amounts of the GS fractions were found in the HFZ and LFZ, correlating only with GLI fraction (r = 0.78) and not with HMW and LMW-GS (Appendix A).

Figure 3 represents the map obtained classifying NIR-measured protein content at two levels (>13% green and ≤13% yellow) and used to carry out the differential harvest. The area with protein content >13% showed a good overlapping with that presenting the higher amount of the total gluten-extractable protein fraction in wholemeal semolina (Figure 1). In particular, this area showed a higher amount of LMW-GS and HMW-GS with respect to the other area selected by NIR with ≤13% protein (Figure 2), corresponding, for the majority, to the MFZ, in which a positive correlation was found between total GS and LMW-GS and the total protein content (Figure 2B,C and Appendix A). This suggests that the >13% protein fraction of grains selected in the field could be more suitable for the production of pasta with high-quality characteristics [7,8]. Conversely, the area with protein content <13%, mainly LFZ and HFZ, overlaps with that showing the lowest content of total gluten-extractable proteins (Figure 1) with a prevalence of the GLI fraction over GS fractions, a subsequent decrease in GS/GLI ratio and a lower correlation of LMW-GS with total protein content (Figure 2, Appendix A). It is well known that a good-quality pasta is determined not only by the high total protein content in the seed but also by the high ratio between GS and GLI [7,8]. Thus, to verify that the differentiated segregation of grains with different protein contents at harvest in the field can lead to products with different characteristics, the quality of the dough obtained from the semolina of the two batches of grains and the corresponding dry pasta was evaluated.

### 2.2. Empiric Rheology

The rheological behavior of semolina samples is reported in Table 1. Samples considered in this study significantly differed for extensibility, with semolina with low protein (≤13%) being more extensible than high-protein semolina (>13%), resulting in a lower P/L value (Table 1). On the other hand, no significant differences were observed in either dough tenacity (P) or strength (W).

The alveograph indices are widely used in the pasta sector for defining the gluten quality of semolina. For instance, considering the ratio between tenacity and extensibility, high values (i.e., >1) are associated with strong gluten, while low values (i.e., <0.5) are associated with weak gluten that is not suitable for pasta production. Regarding W, high values are associated with the formation of a strong network able to retain starch granules during cooking [4]. Thus, W is able to successfully predict the cooking quality of pasta. Specifically, W is negatively correlated with HMW-GS, whereas extensibility (which is not required for pasta production) is negatively correlated with LMW-GS [16].

Gluten aggregation properties of semolina samples are reported in Figure 4 and the related indices in Table 1. When semolina and water are mixed (1:1 ratio), the consistency of the sample increases up to a peak that corresponds to the maximum gluten aggregation. The time of maximum consistency (i.e., of maximum gluten aggregation) is called peak maximum time. After this point, the consistency decreases following the breaking of the gluten network due to intense mechanical action (i.e., the paddle rotates at about 3000 rpm).

Semolina from areas with high protein content (>13%) exhibited lower maximum torque, longer peak maximum time (i.e., the time required to achieve the maximum torque) and higher total energy (i.e., the area under the curve from the beginning of the test up to 15 s after maximum torque) than samples from areas with low-protein content (<13%), suggesting an overall better quality. High-protein semolina was obtained in the field area, showing >13% of proteins and with a high amount of glutenins (i.e., the sum of HMW-GS and LMW-GS) with respect to gliadins (Figure 2). Generally, low values of peak maximum time and energy indicate poor aggregation properties and, thus, low pasta-making performance [4]. In the case of wholemeal semolina, the GlutoPeak indices (and specifically, maximum torque and energy) were able to successfully differentiate durum wheat cultivars based on the glutenin to gliadin ratio [17]. Finally, as regards the relationship between gluten aggregation properties and pasta quality, it was shown that maximum torque and energy provided information on firmness, stickiness and bulkiness of cooked pasta [5,18].

As expected, the protein content of pasta reflected the protein content in the related semolina (Table 2). Moreover, as protein content decreases, starch content increases, whereas no significant differences were measured in terms of ash, fat and fiber content.

As regards the cooking behavior at the optimal cooking time, both semolina samples resulted in a product with high quality (Table 3). Specifically, water absorption around 100% and cooking loss lower than 4% are indicative of good cooking quality [19]. Comparing the samples, high-protein semolina resulted in a product with higher water absorption and lower cooking loss than semolina with low-protein content, indicating a better cooking quality [19], whereas no differences were detected for any of the textural attributes considered. This was confirmed by the sensory analysis that did not detect any significant differences in firmness, as well as in color, flavor and overall acceptability among the samples that, however, received high scores for all the considered attributes (Figure 5). 

The overall results suggest that the pasta-making process applied might have masked those differences in gluten quality described above (Table 1 and Figure 4), at least when pasta was cooked at the optimal cooking time. Thus, further studies might investigate the cooking behavior of pasta in overcooking.

## 3. Material and Methods

### 3.1. Field Experiment

The location used for this research was a rectangular 13.6 ha field located in Mira, Italy (45°22′ N, 12°8′ E and 2 m a.s.l.) (Figure 6A). Three management zones—low, medium and high fertility (LFZ, HFZ, MFZ)—were delineated in the field by using maps of soil apparent electrical conductivity and other measured soil properties (sand, clay, Olsen P and organic matter), overlaid by using a Fuzzy C-means method [13]. The sand content decreased from ca. 67% in the low-fertility zone to 55% in the high-fertility zone and vice versa the clay content increased from 7.5% to 11%. The field was cultivated with Durum wheat (*Triticum durum* Desf.) var. *Biensur* (Apsovmenti, Voghera, Italy) in the period November 2017–June 2018. Each of the three management zones was equally subdivided into variable-rate application (VRA) and conventional fertilization areas. VRA areas received prescribed rates of N based on Normalized Difference Vegetation Index (NDVI) response [20], while conventional areas received standard uniform rates at the farm manager’s discretion throughout the growing season. In addition to these zones, four field-length by six-meter-wide strips, two nitrogen poor (no applied N) and two nitrogen rich (250 kg/ha), were used for calibration of the NDVI (Figure 6A). The fertilization events were three (28 March 2018, 26 April 2018 and 15 May 2018) of granular urea within the VRA and conventional zones and two (28 March 2018, 26 April 2018) of liquid N for the N-rich strip.

### 3.2. Precision Harvest

A pre-harvest procedure was followed to segregate the grains. It required us to previously evaluate the protein content within the field and its variability and grain quality classes to be separated. Further, 120 points were identified and georeferenced in the VRA zones and in the control strips (Figure 6B) one week before the harvesting. In each of the points, wheat ears were collected in 1 m^2^ plots and shells and grains were analyzed for protein quality (Section 3.3). A near-infrared spectroscopy (NIR) analyser was used for a rapid determination of the protein content. Protein data in the 120 points were then spatially interpolated to build a map of protein content to be used for performing precision harvesting and manufacturing precision pasta. Interpolation was performed using ordinary kriging and its goodness of fit was tested with a leave-one-out cross-validation procedure using ArcGis 10.6.1 (ESRI). According to the protein mean content and distribution, two classes of protein content were identified, lower or higher than 13%. A prescription map for grain segregating was then created by applying a classification algorithm to the protein map (ArcGis 10.6.1, ESRI).

### 3.3. Grain Sampling and Gluten Protein Quantification

Thirty-five g of whole grains was milled with Knifetec 1095 (Foss, Hillerød, Denmark) to obtain a fine powder of refined semolina.

Gluten protein fractions (i.e., GLI, HMW-GS and LMW-GS) were extracted with the protocol described by [9]. Thirty mg of semolina was subjected to extraction with 1.5 mL of 550 mL L^−1^ propan-2-ol for 20 min with continuous mixing at 65 °C, followed by centrifugation at 10,000× *g* for 5 min. This step was repeated three times and the supernatants were combined and dried in a vacuum centrifuge to obtain the protein GLI fraction. The remaining pellet containing the GS fractions was suspended in a 400 μL solution of 550 mL L^−1^ propan-2-ol, 0.08 mol L^−1^ tris(hydroxymethyl)aminomethane hydrochloric acid (Tris–HCl, pH 8.3) and 10 g L^−1^ 1,4-dithiothreitol (DTT, as reducing agent) and incubated for 30 min at 60 °C with continuous mixing. After centrifugation at 14,000× *g* for 5 min, the supernatant containing the HMW-GS and LMW-GS fractions was transferred to a new tube. To precipitate HMW-GS, acetone was added to obtain a final concentration of 400 mL L^−1^, which was then centrifuged at 14 000× *g* for 10 min. The LMW-GS fraction was precipitated in the remaining supernatant by adding acetone to obtain a final concentration of 800 mL L^−1^ and this was then centrifuged at 10,000× *g* for 10 min. The GS fractions and gliadins were dissolved in 500 mL L^−1^ acetonitrile (ACN) with 1 mL L^−1^ trifluoroacetic acid (TFA); relative quantification was determined by colorimetric Bradford assay (Bio-Rad, Hercules, CA, USA). Three technical replicates were performed for each sample. Data on gliadins, HMW-GS and LMW-GS fractions were summed to obtain for each sample the total amount of gluten proteins extracted.

Data on protein and gluten protein fractions (both the total amounts of gluten fraction and the different classes) were spatially interpolated by using ordinary kriging and its goodness of fit was tested with a leave-one-out cross-validation procedure (ArcGis 10.6.1, ESRI).

### 3.4. Semolina Characterization

Extensional properties were investigated by using the Alveograph (Chopin, Villeneuve La Garenne Cedex, France) device, following [21].

The gluten aggregation kinetics of durum wheat were investigated by means of the GlutoPeak (Brabender GmbH & Co. KG, Duisburg, Germany) device, following the method reported by Grassi et al. [17]. Briefly, 9 g of sample was added to 9 g of water (adjusting the quantity of sample on 14 g 100 g^−1^ moisture basis) and mixed at 2750 rpm, keeping the bowl temperature at 36 °C.

Results are reported as mean ± standard deviation of three replicates.

### 3.5. Pasta Production

Pasta samples (Appendix A) were produced at a 30% hydration level in a lab-scale extruder (20 kg h^−1^; MAC 30, Italpast, Parma, Italy), keeping the extrusion temperature at 50 °C. The samples were dried in an experimental drying cell under a low-temperature drying cycle (60 °C max; 17 h).

### 3.6. Proximate Composition of Pasta Samples

The chemical composition was performed according to official methods of the American Association of Cereal Chemists [22] for moisture, protein, starch, total fiber, fat and ash. All determinations were carried out on three individual measurements.

### 3.7. Pasta-Cooking Behaviour and Texture Determination

Pasta optimal cooking time (OCT) was determined applying the approved methods 66–51.01 of the American Association of Cereal Chemists [22]. Cooking loss (CL g solids/100 g of dry pasta) was measured by evaporating the cooking water to dryness overnight in a forced-air oven at 110 °C. Water absorption (WA) was measured as the weight increase in pasta after cooking and expressed as percent weight gain with respect to the weight of uncooked pasta. All determinations were carried out on three individual cooking trials.

The textural properties of cooked pasta were performed using a TA.XT. plus instrument Texture Analyser (Stable Micro Systems, Godalming, UK) supported with a 5 kg load cell. A single piece of pasta (length of about 2.4 cm and height of about 12.5 mm) was put on the lower plate and the rectangular probe (30 mm × 50 mm) was moved down onto the pasta surface (test speed of 2 mm s^−1^ and percentage deformation of 95%). Firmness was measured as the maximum peak force required to compress the cooked pasta sample (N), whereas adhesiveness was measured as the maximum peak force curve (g/s) required to withdraw the probe from the surface of the same samples. Data are mean of ten measurements from three different cooking replications

### 3.8. Sensory Evaluation

To test the acceptability of pasta, a sensory evaluation was carried out by 15 testers with experience in food evaluation (9 women and 6 men, aged from 22 to 40 years). The two pasta samples were cooked at the optimal cooking time, drained and served on plastic plates, labelled with random 2-digit codes. Panelists were asked to indicate color, flavor, firmness and overall quality of the products using a ranking test. To this end, a five-point scale, with “1” representing low intensity and “5” high intensity, was used to quantify each attribute. Mean sensory scores from the 15 panel members were calculated for each sample for attributes determined.

### 3.9. Statistical Analyses

Comparison between samples was carried out using Statgraphics Plus 5.1 (StatPoint Inc., Warrenton, VA, USA) and applying a *t*-test. Correlation analyses were performed with SAS/STAT Data Analysis Software (SAS Institute, Cary, NC, USA) considering as variables: total amount of proteins detected by NIR, total extractable-gluten proteins, HMW-GS, LMW-GS, GLI, Total_GS, GS/GLI ratio and Different fertility zones (HFZ, MFZ, LFZ).

## 4. Conclusions

In the experiments carried out in this work, the variability in grain protein content in different field zones assessed by constructing maps of the quantity and quality of gluten proteins in the field was significant enough to justify the segregation of grain during harvest.

The adopted approach was effective in segregating durum wheat plants with different rheological properties that can potentially behave differently during the pasta-making process. Indeed, the pasta produced showed differences in water absorption and cooking loss, indicating different cooking behavior. It cannot be excluded that the scale up of the pasta-making process—as well as the increase in cooking time—would result in increasing such differences, above all, in relation to textural and sensory properties.

Our data demonstrate that segregating grains during harvesting allows one to obtain raw materials with different features and could be a potential asset for a short supply chain, aiming to improve traceability and environmental and economic sustainability.

However, extending this approach to the industrial scale poses a limitation in guaranteeing an adequate tonnage of homogeneous grain stocks. A multi-field or multi-farm approach is imperative to fulfil the grains quantity–quality standards requested by the industrial pasta manufacturing chain.

## Figures and Tables

**Figure 1 plants-11-03149-f001:**
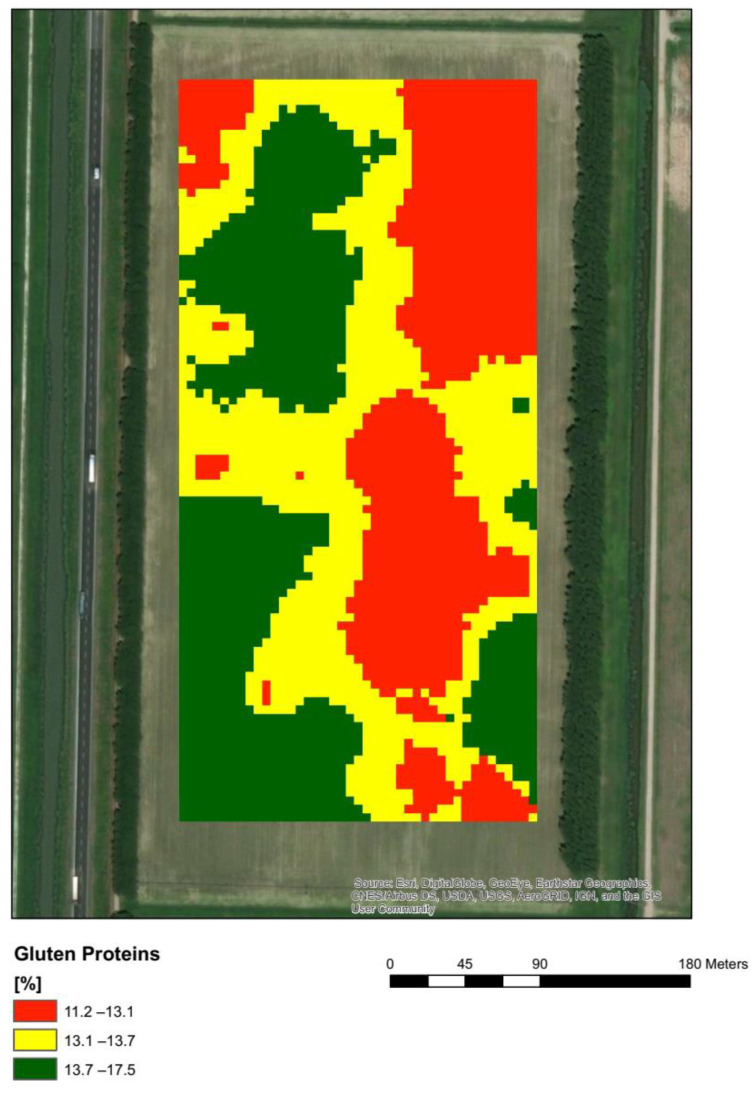
Map of total protein content distribution in the field according to the data of gluten protein extractions.

**Figure 2 plants-11-03149-f002:**
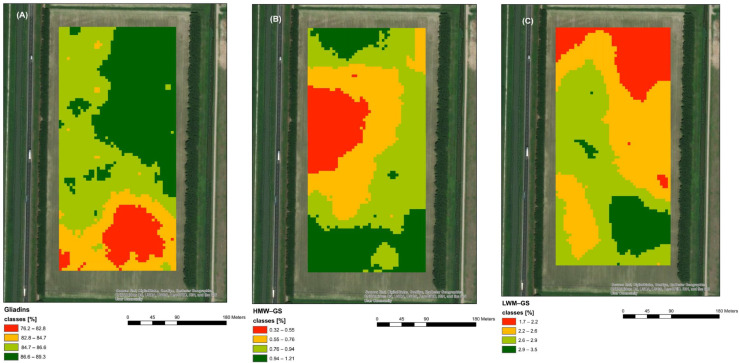
Maps of the contents of the different classes of gluten proteins in the field: (**A**) gliadins; (**B**) high-molecular-weight glutenins (HMW-GS); (**C**) low-molecular-weight glutenins (LMW-GS).

**Figure 3 plants-11-03149-f003:**
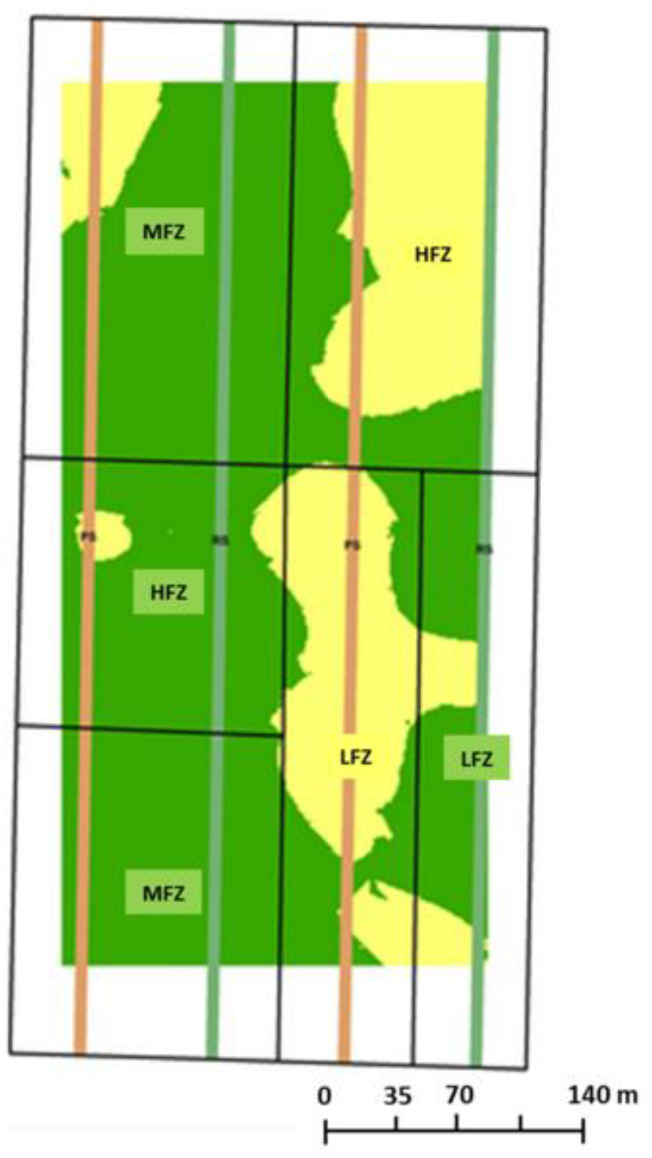
Map of protein content in the field (detected by NIR) for precision harvesting (green areas >13%; yellow areas ≤13%). HFW = high fertility zone; MFW = medium fertility zone; LMW = low fertility zone.

**Figure 4 plants-11-03149-f004:**
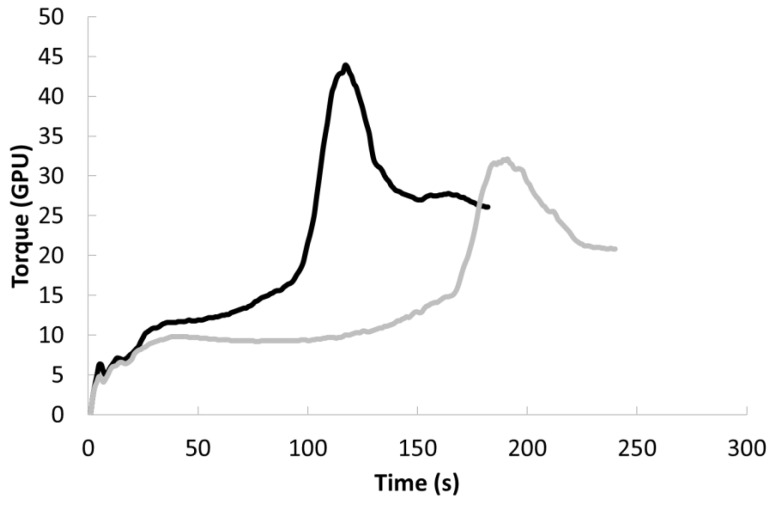
Gluten aggregation properties of semolina samples. Black line: <13%; grey line: >13%. GPU, GlutoPeak Unit.

**Figure 5 plants-11-03149-f005:**
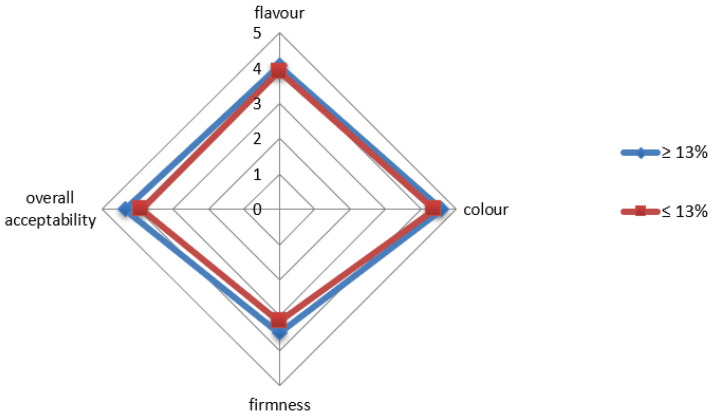
Sensory properties of pasta samples. Each attribute was assessed on a 5-point scale from 1, low intensity, to 5, high intensity.

**Figure 6 plants-11-03149-f006:**
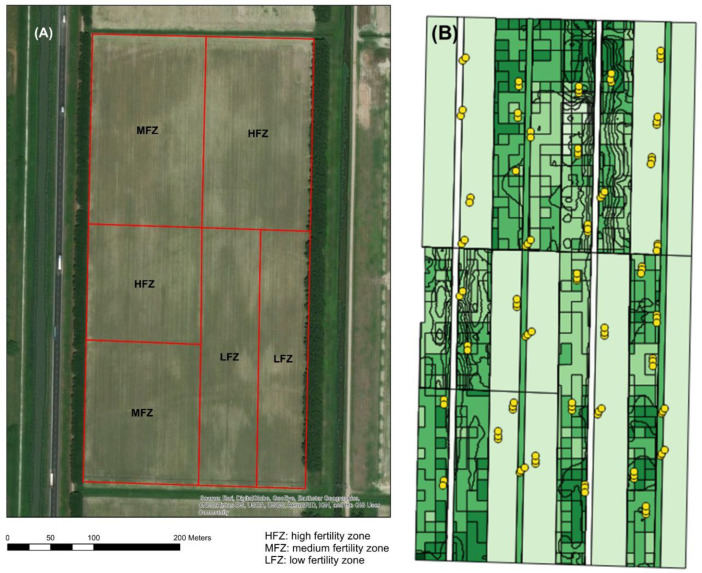
(**A**) Location of study site (and different fertility zones; (**B**) 120 sampling points indicated with yellow dots. HFW = high fertility zone; MFW = medium fertility zone; LMW = low fertility zone.

**Table 1 plants-11-03149-t001:** Indices obtained by the Alveograph and Glutopeak tests.

		Low Protein Semolina (≤13%)	High Protein Semolina (>13%)
Alveograph test	Tenacity (P; mmH_2_O)	41 ± 3 n.s.	41 ± 3
	Extesibility (L;mm)	139 ± 12 **	107 ± 28
	P/L	0.29 ± 0.05 **	0.41 ± 0.11
	Strenght (W; ×10^−4^ J)	139 ± 8 n.s.	130 ± 22
GlutoPeak test	Peak maximum time (s)	118 ± 1 ***	190 ± 2
	Maximum torque (GPU)	44 ± 1 ***	32 ± 0.1
	Total energy (GPE)	2355 ± 39 **	2713 ± 1

GPE, GlutoPeak Equivalent; GPU, GlutoPeak Unit. ** (*p* < 0.01); *** (*p* < 0.001); n.s. (no significant differences); *t*-Test.

**Table 2 plants-11-03149-t002:** Chemical composition of pasta samples. Data are expressed on 100 parts of dry matter. Data are means of 3 replicates. ** (*p* < 0.01); n.s. (no significant differences); *t*-Test.

Pasta	Protein	Ash	Fat	Fiber	Starch
High protein semolina (>13%)	14.21 **	0.79 n.s.	1.80 n.s.	3.07 n.s.	78.86 **
Low protein semolina (≤13%)	12.60	0.75	1.90	2.97	80.17

**Table 3 plants-11-03149-t003:** Optimal cooking time (OCT), water absorption (WA), cooking loss (CL), firmness and adhesiveness of cooked pasta. * (*p* < 0.05); n.s. (no significant differences); *t*-Test.

Pasta	OCT (min)	WA (%)	CL (%)	Firmness (N)	Adhesiveness (g s^−1^)
High protein semolina (>13%)	8	109.4 ± 5.3 *	3.6 ± 0.6 *	16.12 ± 1.46 n.s.	−0.045 ± 0.016 n.s.
Low protein semolina (≤13%)	7.30	96.7 ± 5.4	4.4 ± 0.3	14.76 ± 2.04	−0.059 ± 0.017

## Data Availability

Not applicable.

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
