# Peer review of "Protein Maps for Durum Wheat Precision Harvest and Pasta Production"

_plants, 2022, doi:10.3390/plants11223149_

Round 1
Reviewer 1 Report
It is well known that durum wheat is the highest quality raw material for the pasta industry. The quality depends primarily on the quantity and technological properties of the wheat protein. In order to monitor the quality of wheat (and thus the quantity and properties of proteins), various rheological methods and devices are used, such as the Alveograph and GlutoPeak, which were used in this study. The gluten proteins (high molecular weight glutenin and low molecular weight glutenin, and gliadin) directly affect the rheological properties of wheat. The amount of gluten protein is a genetically determined trait, but is influenced by climatic conditions, soil, fertilization, and the ratio between the different gluten fractions.
In this article, after sampling and measuring the amount of protein in the field and analyzing it in the laboratory, protein maps were prepared regarding the quantity and quality of gluten proteins. Moreover, the technological quality of durum wheat semolina obtained from the grains of a certain part of the field was studied, and pasta samples were prepared from this semolina. In addition, cooking tests and sensory evaluation of each pasta sample were performed.
Except from minor technical inadequacies (e.g., literature references numbered 16, 17, and 18 lack the bolded year) and minor corrections in two tables, I have not found any major deficiencies in the paper.
I ask the authors to reconsider the individual designations in Table 1 so as not to confuse the reader. Indeed, they have indicated below the table the meaning of the asterisks (*, ** and ***). Considering the results, it turned out that none of the numerical values in the given table has significant differences p < 0.05, i.e. an asterisk (*). However, an asterisk (*) appears in the table in meaning of multiplication (*10-4 J). The difference between * (p < 0.05) and *10-4 J is undeniable, but she would still suggest that it be corrected in some way.
In Table 3, there is an "a" next to the numerical value 3.6 ± 0.6, probably by mistake, instead of an asterisk; please reconcile this with the information in the description.
In general, the article is well organized and comprehensively described, contains all necessary components (Introduction, Results and Discussion, clearly explained methodology, etc.), and is written in an understandable manner. In addition, the literature is adequate, and the authors answered all the questions they planned to answer. Considering all these points, I believe that the article should be accepted for publication.
Author Response
Ref#1
It is well known that durum wheat is the highest quality raw material for the pasta industry. The quality depends primarily on the quantity and technological properties of the wheat protein. In order to monitor the quality of wheat (and thus the quantity and properties of proteins), various rheological methods and devices are used, such as the Alveograph and GlutoPeak, which were used in this study. The gluten proteins (high molecular weight glutenin and low molecular weight glutenin, and gliadin) directly affect the rheological properties of wheat. The amount of gluten protein is a genetically determined trait, but is influenced by climatic conditions, soil, fertilization, and the ratio between the different gluten fractions.
In this article, after sampling and measuring the amount of protein in the field and analyzing it in the laboratory, protein maps were prepared regarding the quantity and quality of gluten proteins. Moreover, the technological quality of durum wheat semolina obtained from the grains of a certain part of the field was studied, and pasta samples were prepared from this semolina. In addition, cooking tests and sensory evaluation of each pasta sample were performed.
Except from minor technical inadequacies (e.g., literature references numbered 16, 17, and 18 lack the bolded year) and minor corrections in two tables, I have not found any major deficiencies in the paper.
I ask the authors to reconsider the individual designations in Table 1 so as not to confuse the reader. Indeed, they have indicated below the table the meaning of the asterisks (*, ** and ***). Considering the results, it turned out that none of the numerical values in the given table has significant differences p < 0.05, i.e. an asterisk (*). However, an asterisk (*) appears in the table in meaning of multiplication (*10-4 J). The difference between * (p < 0.05) and *10-4 J is undeniable, but she would still suggest that it be corrected in some way.
In Table 3, there is an "a" next to the numerical value 3.6 ± 0.6, probably by mistake, instead of an asterisk; please reconcile this with the information in the description.
In general, the article is well organized and comprehensively described, contains all necessary components (Introduction, Results and Discussion, clearly explained methodology, etc.), and is written in an understandable manner. In addition, the literature is adequate, and the authors answered all the questions they planned to answer. Considering all these points, I believe that the article should be accepted for publication.
Answer to reviewer#1 We thank the reviewer for the interest in this paper and for the valuable suggestions. We checked all the references at the end of the manuscript, and we formatted them according to Plants requirements. We also modified Table 1, substituting the asterisk meaning multiplication with x (X 10-4 J) to avoid misunderstanding. We also substitute “a” in Table 3 with an asterisk.
Reviewer 2 Report
The manuscript is related to the determination of protein maps for the precise harvesting of durum wheat and the production of pasta in the context of the quality of the raw material (grain). In work based on the grain quality spatial distribution, two distinct areas with protein content lower or higher than 13% were delineated and harvested selectively. The rheological properties of semolina samples obtained from those areas were evaluated. Furthermore, dry pasta was produced and characterized for its cooking behaviour and sensory characteristics. The title of the thesis is communicative and compatible with the content of the reviewed dissertation. It is fully justified, both for cognitive and practical reasons. The individual chapters of the work overlap and constitute a compendium of knowledge on the above subject. The literature information and the conducted experiments are interesting and are in line with the current trends in agriculture (producing an attractive food assortment in the context of very demanding and conscious recipients-consumers). Chapter Results and discussion presented well in terms of content and logically using appropriate statistical methods. The research methods used are described in detail and in detail in the Material and Methods chapter. The Conclusions is a consequence of discussing the results and their discussion. I have no comments. The work is supported by well-chosen literature, but a few of them are not recent (older than 5 years). Please update them. Perhaps it would be good to enrich the work with a few more literature. Besides, please pay attention to the font of the text, commas, spaces and dashes in individual literature items.
Author Response
Ref#2
The manuscript is related to the determination of protein maps for the precise harvesting of durum wheat and the production of pasta in the context of the quality of the raw material (grain). In work based on the grain quality spatial distribution, two distinct areas with protein content lower or higher than 13% were delineated and harvested selectively. The rheological properties of semolina samples obtained from those areas were evaluated. Furthermore, dry pasta was produced and characterized for its cooking behaviour and sensory characteristics. The title of the thesis is communicative and compatible with the content of the reviewed dissertation. It is fully justified, both for cognitive and practical reasons. The individual chapters of the work overlap and constitute a compendium of knowledge on the above subject. The literature information and the conducted experiments are interesting and are in line with the current trends in agriculture (producing an attractive food assortment in the context of very demanding and conscious recipients-consumers). Chapter Results and discussion presented well in terms of content and logically using appropriate statistical methods. The research methods used are described in detail and in detail in the Material and Methods chapter. The Conclusions is a consequence of discussing the results and their discussion. I have no comments. The work is supported by well-chosen literature, but a few of them are not recent (older than 5 years). Please update them. Perhaps it would be good to enrich the work with a few more literature. Besides, please pay attention to the font of the text, commas, spaces and dashes in individual literature items.
Answer to reviewer#2 We thank the reviewer for appreciating our work. We checked all the references and we substituted when possible the older ones with more recent references. We also insert new references in the introduction section according also to Ref#3. In addition, we checked all the text and the references for font, commes, spaces and dashes.
Reviewer 3 Report
The manuscript presents the influence of durum wheat cultivation and harvest conditions on the pasta quality. The subject is relatively simple, the paper is well structured and written, but it is not very complex.
I suggest to perform a PCA or correlations to strengthen the statistics of the paper.
The introduction presents enough the background of the study. The results are discussed, and the materials and methods are enough described.
L35-42: When numbers are given, a citation of the source of information is needed. Please add the corresponding citations after each piece of information containing numbers.
The limitations and the future perspectives should be mentioned in the Conclusion section.
Author Response
Ref#3
The manuscript presents the influence of durum wheat cultivation and harvest conditions on the pasta quality. The subject is relatively simple, the paper is well structured and written, but it is not very complex.
Answer to reviewer#3 We thank the reviewer for valuable suggestions
I suggest to perform a PCA or correlations to strengthen the statistics of the paper.
Answer to reviewer#3: We performed correlation analyses as requested between total protein contents and gluten proteins extracted and the different fertility zones to explain better the data of the protein maps. At this purpose the correlation matrix is inserted as supplementary table 1.
The introduction presents enough the background of the study. The results are discussed, and the materials and methods are enough described.
L35-42: When numbers are given, a citation of the source of information is needed. Please add the corresponding citations after each piece of information containing numbers.
Answer: The reviewer was right and we thank the reviewer for the suggestion. We inserted citations after the information containg numbers in the Introduction section
The limitations and the future perspectives should be mentioned in the Conclusion section.
Answer : We thank the reviewer for the suggestion. We implement the Conclusion section with a phrase about the limitation and the future perspectives of the work
Round 2
Reviewer 3 Report
The authors answered the tasks. However, I think that more discussion on the correlations are needed (complete L140).
The tables in the supplementary material should be completed: please mention below which means color values (I guess there are the significant values), please explain the abbreviations below the table and keep only 2 or maximum 3 decimals for all values.
Author Response
Ref #3
The authors answered the tasks. However, I think that more discussion on the correlations are needed (complete L140).
Answer: We tried to better explain the concept at L140, referring also to the correlations in Suppl. Table 1.
The tables in the supplementary material should be completed: please mention below which means color values (I guess there are the significant values), please explain the abbreviations below the table and keep only 2 or maximum 3 decimals for all values.
Answer: We modify the Supplementary table according to reviewer suggestions